# Does Geographical Discrimination Exist in Online Lending in China: An Empirical Study Based on Chinese Loan Platform Renren

**Tianlei Pi \*, Yaosen Liu and Jiahui Song**

School of Economics and Business Administration, Chongqing University, Chongqing 400030, China; 201702021001@cqu.edu.cn (Y.L.); 1830281@tongji.edu.cn (J.S.)
* Correspondence: pitianlei@cqu.edu.cn; Tel.: +86-136-5768-0656

**Abstract:** Background: Online lending has developed rapidly in China in recent years, into a typical Internet financial model. China's online lending related issues have received widespread attention from scholars. Methods: This study used 396,634 order data-points (935,037 original order data-points) from the Renren Loan website since its inception in January 2017. We used ordinary least squares (OLS) regression to study the problem of geographical discrimination in online lending in China, and we conducted two robust tests. Results: Studies have shown that significant geographical discrimination exists in China's online lending market. From the perspective of the lender, different investment intentions exist for borrowers from various regions, thereby leading to variations in the success rates of loans. From the perspective of the borrower, the belief exists that borrowers from different regions will have varying interest rates because of the effect of geographical discrimination. Conclusion: We believe that geographical discrimination is due to the effects of the economic, financial, educational, and ethnic conditions of the borrower's location on willingness to invest and the success rate of borrowing. However, borrowers' self-discrimination is primarily related to economic and ethnic differences among provinces.

**Keywords:** online lending; geographical discrimination; success rate of borrowing; borrowing rate

**JEL Classification:** D80; H70; J71

## 1. Introduction

Internet lending is a financial model that has emerged in China in recent years along with the development of Internet finance (Chen et al. 2013). Online lending plays an important role in eliminating middlemen, reducing transaction costs, and increasing the interests of borrowers and lenders (Chi et al. 2019), thereby providing a powerful complement to the traditional financial industry (Ma et al. 2018). Online lending is rapidly developing in China because of its low financing threshold, convenient procedures, wide coverage, and efficient allocation of funds (Pi and Zhao 2014; Lin et al. 2018). After the explosive growth, a large number of industry crises have emerged, such as difficulty in paying and competing, which have elicited widespread concern in the academic circles and the society.

Table 1 describes the development of China's online lending industry, and Figure 1a shows the growth of China's online lending transaction volume. China's online lending industry is undergoing a period of adjustment after experiencing a sharp rise. Figure 1b summarizes the number of downtime and problem platforms in the online lending industry from 2015 to 2018. This figure shows that China's online lending industry remains immature, and risk pressure is still relatively large. This

information indicates the difficulties that China's online lending platform is currently experiencing. However, it also shows that China's current regulation on online lending is becoming increasingly mature, and online lending is gradually embarking on a path toward healthy development.

**Table 1.** Basic situation and stage division of China's peer-to-peer (P2P) development.

| Time | Stage | Characteristics |
|------|-------|-----------------|
| 2007–2011 | Start-up period | The mode of copying foreign platforms |
| 2011–2012 | Expansion period | Rapid increase in quantity |
| 2013–2014 | Risk outbreak | Problem online loan company increased |
| 2014–2017 | Policy adjustment period | Government strengthens supervision |
| 2017–now | Remodeling period | Remodeling business model |

Source: author collection judgment.

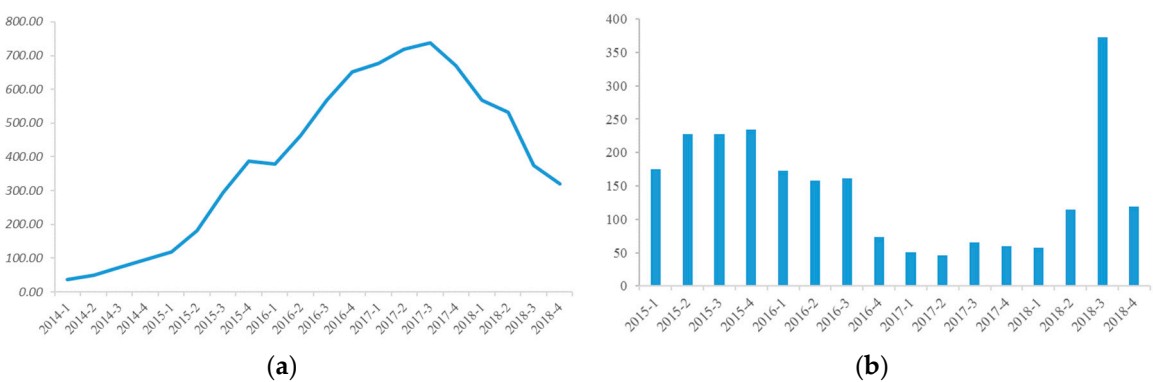

(**a**)                                    (**b**)

**Figure 1.** China's online lending transaction volume (**a**) and number of problematic platforms (**b**).

Similarly to the ups and downs of China's online lending industry, the development of online lending in China's provinces also differs. Figure 2 summarizes the distribution of network lending platforms that are operating normally in each province in 2018. By the end of 2018, the top four regions with the highest numbers of online lending platforms that were operating normally were Guangdong (236), Beijing (211), Shanghai (114), and Zhejiang (79). The number of platforms in the four provinces accounts for 61.84% of the total number of online lending platforms in China.

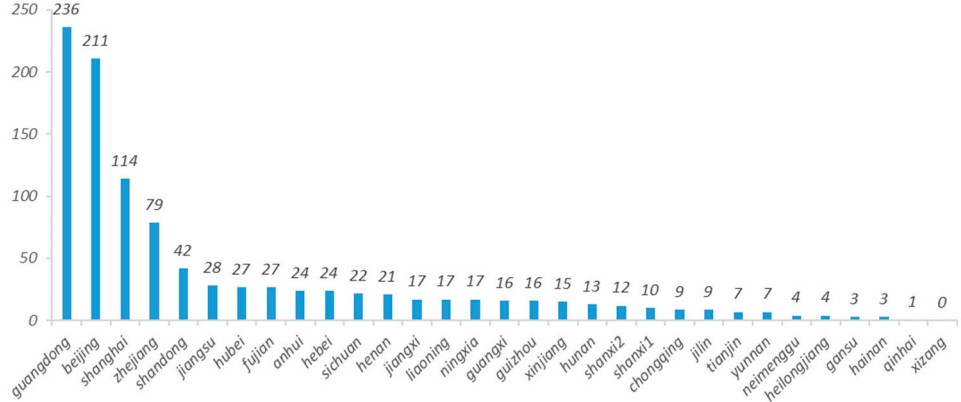

**Figure 2.** Number of normal operating online lending platforms in all provinces of China at the end of 2018.

The uneven distribution of the number of online lending platforms in China intuitively illustrates discrimination in China's online lending. In consideration of this situation, the current study uses the loan data of "Renren Dai", one of the earliest peer-to-peer (P2P) network lending platforms in China, to

analyze the success rate of borrowing among different regions and the interest rate of borrowing from the perspective of lenders and borrowers. This study also explores whether geographical discrimination exists in the P2P network lending market.

In addition, we analyze the development of geographical discrimination in the online lending market from four perspectives: economy, finance, education, and ethnicity. The results show that significant geographical discrimination exists in the online lending market in China. From the perspective of the lender, different investment intentions exist for borrowers from various regions, thereby leading to variations in the success rate of borrowings. From the perspective of the borrower, the belief exists that borrowers from different regions have varying borrowing rates because of the effect of geographical discrimination. For lenders, the economic, financial, educational, and ethnic conditions of the borrower's location affect their willingness to invest, thereby resulting in a difference in the success rate of borrowing. For borrowers, self-discrimination is primarily related to economic and ethnic differences among provinces.

This study makes useful contributions from the following perspectives.

First, we enrich the research scope in the field of online lending discrimination. Existing research on online lending focuses on discrimination based on personal characteristics, such as race, age, appearance, and gender. Ravina (2012) found that personal characteristics, such as appearance, race, and age, influence the decision of the lender. Duarte et al. (2009) reported evident discrimination against Asian, Hispanic, young, and overweight individuals in the US P2P lending market. Chen et al. (2018) determined that borrowers with a bachelor's degree had higher borrowing rates than borrowers with associate degrees, and that female borrowers had higher education costs than males. Carlos (2018) noted that female lenders have higher quality loans. Pope and Sydnor (2011) showed that blacks may be less present than whites in the online lending market, and their lending rates are higher than those of whites. In addition, the elderly and overweight groups are treated unfairly. Investors are highly inclined to borrow money for women and individuals with a military background. Li et al. (2013) pointed out that age discrimination and identity discrimination exist in the P2P network lending market in China. Chen and Ye (2016) showed that irrational preference and discrimination are prevalent against female borrowers in China's Internet financial market, with single female borrowers being seriously discriminated. Current research on discrimination in P2P network lending is focused on racial, gender, and identity discrimination. Only a few studies have focused on geographical discrimination in online lending, and no exhaustive inquiry has been made into the causes and performance rules of geographical discrimination. The current study will supplement and improve research on geographical discrimination in online lending and explore its causes and inherent laws.

Second, this study will supplement research in the field of online lending in China. At present, most studies on China's P2P online lending focus on operational models and risk supervision, whereas only a few studies focus on discrimination in the Internet financial lending market. Moreover, analyses that concentrate at the macro level are rare. Liu and Xia (2018) analyzed the impacts of policies on the online lending industry by constructing a game analysis. Yang et al. (2018) explored the risk causes of online lending on the basis of game theory and an information asymmetry model of information economics. Few studies have also conducted analyses at the micro level. Jiang and Zhou (2016) explored the income discrimination phenomenon in the online loan market by dividing a group into high and low income. However, the difference in provincial discrimination has been minimally discussed. Therefore, the current study is an important supplement to the research on provincial and regional discrimination in the field of online lending.

Lastly, our research also provides a reference for the healthy development of China's online lending industry. It is useful for addressing the unbalanced geographical development of P2P network lending and the considerable difference in transaction volume among regions. This study helps Internet financial platforms to rationally plan their strategies, develop effective financing channels, and establish customer resources in a targeted manner. Research support is provided for the development direction and strategic planning of the financial platform.

The remainder of this paper is organized follows. The second part presents the theoretical analysis and research hypothesis of the phenomenon of geographical discrimination in online lending. The third part describes the data situation, variable settings, and empirical model setting. The fourth part analyzes the existence of geographical discrimination in online lending from the perspectives of lenders and borrowers. It also investigates the causes of regional discrimination in online lending from the aspects of economy, finance, education, and ethnicity. The fifth part presents the conclusions drawn from the research.

## 2. Theoretical Analysis and Proposed Hypotheses

A large degree of information asymmetry exists in online lending (Feng et al. 2019; Caldieraro et al. 2018), and lenders must use limited, readily available, objective information and expertise to assess the risk of loan applications. Information can influence lenders' loan decisions to a considerable extent, thereby making lenders' decisions highly subjective (Jin et al. 2019). Readily available information, such as a borrower's income, age, and education, exerts a significant impact on a lender's risk of default (Hu et al. 2019). Geographic information is frequently the most easily accessible data in Chinese online lending transactions. Therefore, geographical information is important to the lender and will affect his/her loan decision.

Geographical discrimination in China has numerous causes, such as discrimination in economically underdeveloped, rural, and low-income areas. An imbalance in the income level has existed for a long time among provinces in China, and the incomes of residents in different regions exhibit considerable differences. Such discrepancies may lead to geographical discrimination in the relationship between loans and unfair treatment caused by the aforementioned differences. That is, borrowers from low-income provinces may be discriminated against, whereas borrowers from high-income provinces may be more favored.

Mollick (2014) found that the success rate of the popular US crowdfunding website, Kickstarter, varies from region to region. Lin and Viswanathan (2015) found the existence of local bias in online financial markets, where transactions are frequently likely to occur among people in the same country or region. Lucia and Mattarocci (2018) found that the regional population income of UK's P2P loan service is lower than the national average, and real estate prices are below average. These studies show that regional information affects online lending. In a loan relationship, the borrower's income level, particularly the disposable income level, will directly affect his/her ability to repay and default. Differences in income among various regions in China have existed for a long time, and the incomes of residents from various regions exhibit considerable differences. Such discrepancies can lead to geographical discrimination between loans and the resulting unfair treatment. That is, borrowers from low-income areas may be discriminated against.

**Hypothesis 1.** *A geographical phenomenon of lender's discrimination against borrowers exists in the online lending industry in China.*

In the study of online lending discrimination, apart from the information discrimination of the lender against the borrower, self-discrimination of the borrower also exists. Barasinska and Schafer (2014) found that female borrowers have higher borrowing rates than males, primarily because of the self-discrimination of female borrowers. That is, female borrowers attract investors by setting higher interest rates than normal.

We believe that borrowers from economically underdeveloped regions may be discriminated against when considering their geographic location. To increase the success rate of borrowing, borrowers from low-income regions tend to set higher interest rates to attract lenders. By contrast, borrowers from high-income areas tend to set lower interest rates to reduce their borrowing costs, given the informational advantages conveyed by their location.

**Hypothesis 2.** *A geographical phenomenon of self-discrimination among borrowers exists in the Chinese online lending industry.*

If evident geographical discrimination exists in the field of online lending in China, then we will further explore the reasons for such geographical discrimination. Han et al. (2019) believed that financing knowledge is directly related to P2P lending. Barasinska and Schafer (2014) asserted that wisdom is a key factor that affects women's discrimination in borrowing; and different regional economic, educational, and other conditions will affect the development of knowledge in a region. Peng et al. (2016) regarded the economic and education levels, financial ecological environment, and formal financial popularity of provinces as the major reasons that affect geographical discrimination in Internet finance. Huang and Yao (2006) reported that geographical discrimination is typically due to regional cultural differences and economics. It is caused by factors such as uneven development. We further explore the causes of geographical discrimination in online loans from the economic level of each province and from their financial, educational, and ethnic differences. On the basis of such exploration, we propose the following hypotheses:

**Hypothesis 3a (H3a).** *Differences in economic levels among provinces in China cause geographical discrimination.*

**Hypothesis 3b (H3b).** *Fiscal differences among Chinese provinces cause geographical discrimination.*

**Hypothesis 3c (H3c).** *Educational differences among various provinces in China cause geographical discrimination.*

**Hypothesis 3d (H3d).** *Ethnic differences among Chinese provinces cause geographical discrimination.*

## 3. Variables, Data, and Research Design

### 3.1. Data Source and Processing

This study used data from the Renren loan website since its establishment in January 2017. The original order data reach 935,037. The website publishes loan order information, and registered users can view such information for free.

We processed the obtained original order data in the following ways. First, order data disclosed by borrowers with incomplete information were excluded. Second, only order data with credit certification target as order type were retained. The three other types of data (i.e., field certification, organization guarantee, and smart wealth management marks) were excluded because field certification and organization guarantee are concerned with the guarantor's information, whereas a smart financial label is not only a small number, but also closer to being a wealth management product. The credit certification target is more concerned with borrowing than the three other types of labels. The information of people is more in line with our research objectives, and thus, this study selected credit certification target for research. Third, data from Hong Kong, Macau, and Taiwan were excluded because they were insufficient to be representative. Fourth, the data of borrowers who were not between the ages of 18 and 65 years were excluded. Finally, the number of valid data-points obtained was 396,634.

We also used the gross domestic product (GDP) of each province to measure economic differences among regions, local budgetary general expenditures to measure fiscal differences, and education funds to measure educational differences. This study distinguished between ethnic minority groups in autonomous regions and those in other provincial administrative units to measure ethnic differences among regions. The GDP and general budgetary expenditure of the local finance of each province were obtained from the 2016 statistics. The education funds of each province were derived from the 2015 statistics. The data source was the National Bureau of Statistics of China.

*3.2. Model Setting*

3.2.1. Model Setting of Geographical Discrimination

This study examined whether geographical discrimination exists from the perspectives of lenders and/or borrowers by setting the following two models. After controlling other influencing factors, whether a significant difference in the success and interest rates of borrowing exists among different provinces was determined.

First, Model (1) was used to test whether a significant difference in the success rate of borrowing exists among different provinces.

$$success_i = \alpha_0 + \sum \alpha_n \times prov_n + \lambda \times K_i + \mu_i \tag{1}$$

$success_i$ is a dummy variable that indicates whether the *i*-th loan order is successfully borrowed. When the loan order is successful, this variable takes a value of 1; when the loan fails, it takes a value of 0. $prov_n$ indicates the nth province. Among the 31 provinces, Zhejiang was used as the control and 30 dummy variables were set. $K_i$ is the control variable. It includes the information of the order, such as the borrowing rate, the loan amount, the repayment period, and the borrower's personal information (e.g., age, education, income, and other information). $\mu_i$ is the error term.

If the regression coefficient $\alpha\_n$ in Model (1) is significantly combined, then the existence of a significant difference in the success rate of borrowing among different provinces is proven. That is, from the perspective of the lender in network borrowing, geographical discrimination exists against the borrower.

Model (2) was used to test whether a significant difference in borrowing rates exists among different provinces.

$$apr_i = \beta_0 + \sum \beta_n \times prov_n + \delta \times K_i + \mu_i \tag{2}$$

$apr_i$ indicates the borrowing rate of the *i*-th borrowing order.

If the regression coefficient $\beta\_n$ in Model (2) is significantly combined, then the existence of a significant difference in the interest rates of borrowing among different provinces is proven. That is, self-discrimination of borrowers exists from the perspective of borrowers in online lending.

3.2.2. Model Setting for Reasons of Geographical Discrimination

If the aforementioned test results of geographical discrimination prove that geographical discrimination exists in the online lending market, then this study will further explore the causes of geographical discrimination in online lending from four aspects: economy, finance, education, and ethnicity. That is, economics among regions will be analyzed. The impacts of fiscal, educational, and ethnic differences on the success and interest rates of borrowing will also be investigated.

From the perspective of lenders, this study first set Models (3)–(6) to study the impacts of economic, fiscal, educational, and ethnic differences on the success rate of borrowing after controlling other influencing factors.

$$success_i = \alpha_0{}' + \alpha_1{}' \times prov\_gdp_i + \lambda' \times K_i + \mu_i \tag{3}$$

$$success_i = \alpha_0{}'' + \alpha_1{}'' \times prov\_gov_i + \lambda'' \times K_i + \mu_i \tag{4}$$

$$success_i = \alpha_0{}''' + \alpha_1{}''' \times prov\_edu_i + \lambda''' \times K_i + \mu_i \tag{5}$$

$$success_i = \alpha_0{}'''' + \alpha_1{}'''' \times prov\_nat_i + \lambda'''' \times K_i + \mu_i \tag{6}$$

From the borrower's perspective, we set Models (7)–(10) to study the effects of economic, fiscal, educational, and ethnic differences on the interest rate of borrowing after controlling other influencing factors.

$$apr_i = \beta_0{}' + \beta_1{}' \times prov\_gdp_i + \delta' \times K_i + \mu_i \tag{7}$$

$$apr_i = \beta_0'' + \beta_1'' \times prov\_gov_i + \delta'' \times K_i + \mu_i \tag{8}$$

$$apr_i = \beta_0''' + \beta_1''' \times prov\_edu_i + \delta''' \times K_i + \mu_i \tag{9}$$

$$apr_i = \beta_0'''' + \beta_1'''' \times prov\_nat_i + \delta'''' \times K_i + \mu_i \tag{10}$$

Models (3) and (7) were used to study the impacts of economic differences among provinces on the success and interest rates of borrowing. This study used the GDP of each province to measure economic difference among regions. $prov\_gdp_i$ indicates the GDP level of the province where the borrowing order is located. The 31 provinces were divided into five groups in accordance with GDP level. Numbers 1–5 were assigned from low to high. If the regression coefficient $\alpha_1'$ of Model (3) is significantly positive, then it indicates that the higher the GDP, the higher the success rate of borrowing. If $\alpha_1'$ is significantly negative, then it indicates that the higher the GDP, the lower the success rate of borrowing. If $\alpha_1'$ is insignificant, then it indicates that the success rate of borrowing is unaffected by the GDP of the province. If the regression coefficient $\beta_1'$ of Model (7) is significantly positive, then it indicates that the higher the GDP, the higher the borrowing rate set by the borrowers in the province. If $\beta_1'$ is significantly negative, then it indicates that the higher the GDP, the lower the interest rate set by the borrowers in the province. If $\beta_1'$ is insignificant, then it indicates that the interest rate set by the borrowers is unaffected by the GDP of the province.

Models (4) and (8) were used to study the impacts of fiscal differences among different provinces on the success and interest rates of borrowing. We used the local fiscal general budget expenditures of each province to measure financial differences among regions. $prov\_gov_i$ indicates the local fiscal general budget expenditure level of the province where the borrowing order is located. The 31 provinces were divided into five groups in accordance with the local fiscal general budget. The expenditure level was divided into five groups, and each group was assigned a number 1–5 to indicate low to high. If the regression coefficient $\alpha_1''$ of Model (4) is significantly positive, then it indicates that the higher the local budgetary expenditure, the higher the success rate of the province's borrowing. If $\alpha_1''$ is significantly negative, then it indicates that the higher the local budgetary expenditure, the lower the success rate of the province's borrowing. If $\alpha_1''$ is insignificant, then it indicates that the success rate of borrowing is unaffected by the general budgetary expenditure of the local finance in the province. If the regression coefficient $\beta_1''$ of Model (8) is significantly positive, then it indicates that the higher the local budgetary expenditure, the higher the borrowing rate set by the province's borrowers. If $\beta_1''$ is significantly negative, then it indicates that the higher the local budgetary expenditure, the lower the borrowing rate set by the provincial borrowers. If $\beta_1''$ is insignificant, then the interest rate set by the borrower is unaffected by the general budgetary expenditure of the local finance in the province.

Models (5) and (9) were used to study the impacts of educational differences among different provinces on the success and interest rates of borrowing. We use the education funds of each province to measure educational differences among regions. $prov\_edu_i$ indicates the level of educational funding of the province where the borrower is located. The 31 provinces were divided into five groups. Each group was assigned a number from 1 to 5 to indicate low to high in accordance with the level of educational funding. If the regression coefficient $\alpha_1'''$ of Model (5) is significantly positive, then it indicates that the higher the education funds, the higher the success rate of borrowing. If $\alpha_1'''$ is significantly negative, then it indicates that the higher the educational funding of the province, the lower the success rate of borrowing. If $\alpha_1'''$ is insignificant, then it indicates that the success rate of borrowing is unaffected by the education funds of the province. If the regression coefficient $\beta_1'''$ of Model (9) is significantly positive, then it indicates that the higher the education funds of the province, the higher the borrowing rate set by the borrowers in the province. If $\beta_1'''$ is significantly negative, then it indicates that the higher the education funds, the lower the borrowing rate set by the province's borrowers. If $\beta_1'''$ is insignificant, then it indicates that the interest rate set by the borrower is unaffected by the education funds of the province.

Lastly, Models (6) and (10) were used to study the effects of ethnic differences among different provinces on the success and interest rates of borrowing. This study measured ethnic differences among

regions by distinguishing between ethnic minority groups from autonomous regions and groups from other provincial administrative units. $prov\_nat_i$ is the dummy variable of the province in which the borrower indicates the region where the borrowing order is located is a minority autonomous region. The variable takes a value of 1 if the province is a minority autonomous region; otherwise, it takes a value of 0. If the regression coefficient $\alpha_1''''$ of Model (6) is significantly positive, then it indicates that the success rate of borrowing in minority autonomous regions is higher than that in other provincial administrative regions. If $\alpha_1''''$ is significantly negative, then it indicates that the success rate of borrowing in minority autonomous regions is lower than that in other provincial administrative regions. If $\alpha_1''''$ is insignificant, then it indicates that the success rate of borrowing is unaffected by whether the province where the borrower is located is a minority autonomous region. If the regression coefficient $\beta_1''''$ of Model (10) is significantly positive, then it indicates that the interest rate set by the borrowers in minority autonomous regions is higher than that in other provincial administrative regions. If $\beta_1''''$ is significantly negative, then it indicates that the borrowing rate set by the borrowers in minority autonomous regions is lower than that in other provincial administrative regions. If $\beta_1''''$ is insignificant, then it indicates that the borrowing rate set by the borrowers is unaffected by whether the province where the borrower is located is a minority autonomous region.

### 3.3. Definition of Variables

#### 3.3.1. Interpreted Variables

$success_i$: A dummy variable that takes a value of 1 when the borrowing order is successful and a value of 0 when the borrowing order fails. The status of the loan order is either repaid, paid, flown, or overdue. Among these statuses, repayment, advance, paid, and overdue orders are regarded as successful loan orders and are marked.

$apr_i$: The annualized interest rate of the borrowing order set by the borrower to obtain a loan.

#### 3.3.2. Explanatory Variables

$prov_n$: A total of 30 dummy variables were set for the 31 provinces, with Zhejiang as the control.

$prov\_gdp_i$: The GDP level of the province in which the borrower is located. The 31 provinces were divided into five groups in accordance with their GDP ranking. The five groups are as follows: highest GDP group (GDP ranked 1–6), high GDP group (GDP ranked 7–12), medium GDP group (GDP ranked 13–18), low GDP group (GDP ranked 19–24), and lowest GDP group (GDP ranked 25–31). For the borrower's province, the lowest GDP group has a value of 1, the low GDP group has a value of 2, the medium GDP group has a value of 3, the high GDP group has a value of 4, and the highest GDP group has a value of 5.

$prov\_gov_i$: The general budgetary expenditure level of the local finance of the province where the borrower is located. The 31 provinces were partitioned into five groups in accordance with the local fiscal general budget expenditure level. The ranking of the five groups is as follows: highest local finance general budget expenditure group (ranked 1–6), high local finance general budget expenditure group (ranked 7–12), medium local finance general budget expenditure group (ranked 13–18), low local finance general budget expenditure group (ranked 19–24), and lowest local finance general budget expenditure group (ranked 25–31). For the province where the borrower is located, the lowest local finance general budget expenditure group is assigned a value of 1, the low local finance general budget expenditure group is assigned a value of 2, the medium local finance general budget expenditure group is assigned a value of 3, the high local finance general budget expenditure group is assigned a value of 4, and the highest local finance general budget expenditure group is assigned the value of 5.

$prov\_edu_i$: Level of education funds in the province where the borrower is located. The 31 provinces were divided into five groups in accordance by the level of education funds. The ranking of the five groups is as follows: highest education fund group (ranked 1–6), high education fund group (ranked 7–12), medium education fund group (ranked 13–18), low education fund group (ranked

19–24), and lowest education fund group (ranked 25–31). For the borrower's province, the lowest education fund group has a value of 1, the low education fund group has a value of 2, the medium education fund group has a value of 3, the high education funding group has a value of 4, and the highest education fund group has a value of 5.

*prov_nat$_i$*: Dummy variable, takes a value of 1 when the borrower's province is a minority autonomous region; otherwise, it takes a value of 0.

### 3.3.3. Main Control Variables

period: Repayment period of the borrowing order set by the borrower; the period ranges from 1 month to 36 months.

ln_amount: The amount of the borrowing order is taken as the logarithm. In accordance with the provisions of the Renren Credit website, the loan amount should be between 3000 yuan and 1,000,000 yuan. The loan amount should be a multiple of 50.

apr: The annualized interest rate of the borrowing order set by the borrower to obtain the loan. When the explanatory variable is the success rate of borrowing, the borrowing rate is used as a control variable.

credit: Dummy variable. The Renren Credit website divides the credit rating of the borrower into seven grades: AA, A, B, C, D, E, and HR. The HR level indicates that the borrower's credit status is the poorest. When the borrower's credit rating is HR, a value of 1 is taken and the other credit rating is 0.

company: When the size of the company where the borrower is located is less than 10, a value of 1 is taken. When the size of a company is greater than 10 but less than 100, a value of 2 is taken. When the size of a company is greater than 100 but less than 500, a value of 3 is taken. When the size of a company is greater than 500, a value of 4 is taken.

Table 2 summarizes the variables:

**Table 2.** Variable definition and description.

| Variable Type | Variable | Variable Definitions |
|---|---|---|
| Explained variable | success | Take 1 when the loan is successful, and take 0 if the failure is successful. |
| | apr | Annualized interest rate of borrowing orders |
| Explanatory variables | Prov | The province where the borrower is located, with Zhejiang as the control group, set 30 dummy variables |
| | prov_gdp | Assignment from low to high by province GDP level 1–5 |
| | prov_gov | According to the province's local fiscal general budget expenditure level from low to high assignment 1–5 |
| | prov_edu | According to the provincial education funding level from low to high assignment 1–5 |
| | prov_nat | The minority autonomous region takes 1 and the rest takes 0 |
| Control variable | period | Repayment period set by the borrower |
| | ln_amount | The loan amount is logarithm |
| | apr | Annualized interest rate of borrowing orders |
| | credit | Take 1 when the level is HR, otherwise take 0 |
| | age | Borrower's age |
| | marry | Married to take 1, otherwise take 0 |
| | education | According to education from low to high assignment 1–4 |
| | company | Assignment from low to high according to the size of the borrower company 1–4 |
| | income | Assigned from low to high according to the borrower's income level 1–7 |
| | house | Have a property to take 1, otherwise take 0 |
| | car | Have a car take 1, otherwise take 0 |
| | House loan | Have a mortgage to take 1, otherwise take 0 |
| | carloan | Have a car loan to take 1, otherwise take 0 |

## 4. Empirical Analysis of Geographical Discrimination in Online Lending

### 4.1. Descriptive Statistics

The descriptive statistics of the variables are provided in Table 3. We found that the success rate of all online loan orders was 6.77%, and the overall success rate of online lending orders was low. The maximum interest rate set by a borrower was 23.4%, whereas the minimum was 3%. The

average and median were 13.68% and 13%, respectively, which are considerably higher than a bank loan interest rate for the same period. This difference is attributed to the online lending risk premium being higher than the risk of bank lending.

**Table 3.** Descriptive statistics of variables.

| Variable | Number | Mean | Median | Max | Min | Standard |
|---|---|---|---|---|---|---|
| success | 396,634 | 0.0677 | 0 | 1 | 0 | 0.2512 |
| apr | 396,634 | 13.6829 | 13 | 23.4 | 3 | 3.0772 |
| prov_gdp | 396,634 | 3.1127 | 3 | 5 | 1 | 1.1781 |
| prov_gov | 396,634 | 3.8473 | 4 | 5 | 1 | 1.2994 |
| prov_edu | 396,634 | 3.881 | 4 | 5 | 1 | 1.2419 |
| prov_nat | 396,634 | 0.065 | 0 | 1 | 0 | 0.2466 |
| period | 396,634 | 15.9585 | 12 | 36 | 1 | 9.3176 |
| ln_amount | 396,634 | 10.2095 | 10.309 | 13.8155 | 6.9078 | 1.2996 |
| creditrating | 396,634 | 0.9429 | 1 | 1 | 0 | 0.2321 |
| age | 396,634 | 31.1305 | 29 | 65 | 18 | 6.5614 |
| marry | 396,634 | 0.4984 | 0 | 1 | 0 | 0.5 |
| education | 396,634 | 1.8676 | 2 | 4 | 1 | 0.793 |
| company | 396,634 | 2.3936 | 2 | 4 | 1 | 1.0475 |
| income | 396,634 | 3.9143 | 4 | 7 | 1 | 1.1609 |
| house | 396,634 | 0.436 | 0 | 1 | 0 | 0.4959 |
| car | 396,634 | 0.2506 | 0 | 1 | 0 | 0.4334 |
| houseloan | 396,634 | 0.141 | 0 | 1 | 0 | 0.3481 |
| carloan | 396,634 | 0.0573 | 0 | 1 | 0 | 0.2325 |

The average values of provincial GDP, local fiscal general budget expenditure, and educational fund levels are 3.1127, 3.8473, and 3.8810, respectively. The number of online lending orders in regions with higher economic, financial, and educational levels is higher than in regions with lower economic, financial, and education levels. Orders in minority autonomous regions only account for 6.5% of the total orders, thereby indicating that people in minority autonomous regions have relatively minimal exposure to online lending, and online lending is less popular in minority autonomous regions than in other provincial administrative regions.

Table 4 presents the descriptive statistics of the status of loan orders in each province. The province with the highest success rate for borrowing is Zhejiang, where the success rate reaches 8.67%. The province with the lowest borrowing success rate is Chongqing, with only 3.20%, which is less than half of Zhejiang's rate. Such significant differences exist in the success rate of borrowing among provinces.

Table A1 shows a table of correlation coefficients among various explanatory and control variables. The model is unaffected by multicollinearity, as shown by the correlation coefficient between each explanatory and control variables.

### 4.2. Empirical Test of Geographical Discrimination

Table 5 shows that the regression results of Model (1) for the explanatory variables are successful. Model (1) examines whether a significant difference exists in the success rate of borrowing among different provinces under the control of other variables mostly from the perspective of the lender.

As shown in Table 5, the regression coefficients of most provinces are significant at the 1% level in the case in which the other variables are controlled. The lower part of Table 5 shows the difference test results of the success rate of borrowing among the provinces and the difference test results of the success rate between the provinces and the control group. The F values are 9.3961 and 11.0826. Significant differences exist among the regression coefficients of the various provinces. These results show that under the control of other variables, a significant difference exists in the success rate of

borrowing among different provinces. From the perspective of lenders, the phenomenon of lenders' geographical discrimination against borrowers in online lending exists.

**Table 4.** Lending success rates in 31 provinces in China.

| Province | Successful | Failure | Total | Success Rate |
|---|---|---|---|---|
| zhejiang | 2516 | 26,519 | 29,035 | 8.67% |
| jiangsu | 2029 | 22,485 | 24,514 | 8.28% |
| guizhou | 551 | 6338 | 6889 | 8.00% |
| beijing | 1223 | 14,275 | 15,498 | 7.89% |
| gansu | 308 | 3817 | 4125 | 7.47% |
| shanghai | 976 | 12,113 | 13,089 | 7.46% |
| shanxi_2 | 625 | 7979 | 8604 | 7.26% |
| shandong | 2042 | 26,386 | 28,428 | 7.18% |
| qinghai | 63 | 835 | 898 | 7.02% |
| guangxi | 795 | 10,636 | 11,431 | 6.95% |
| guangdong | 4107 | 55,127 | 59,234 | 6.93% |
| yunnan | 582 | 7905 | 8487 | 6.86% |
| henan | 1141 | 15,558 | 16,699 | 6.83% |
| hainan | 208 | 2871 | 3079 | 6.76% |
| tianjin | 227 | 3141 | 3368 | 6.74% |
| neimenggu | 476 | 6735 | 7211 | 6.60% |
| jinlin | 368 | 5288 | 5656 | 6.51% |
| ningxia | 137 | 1980 | 2117 | 6.47% |
| xinjiang | 282 | 4136 | 4418 | 6.38% |
| anhui | 753 | 11,096 | 11,849 | 6.35% |
| heilongjiang | 519 | 7844 | 8363 | 6.21% |
| jiangxi | 569 | 8669 | 9238 | 6.16% |
| fujian | 1355 | 21,324 | 22,679 | 5.97% |
| hebei | 770 | 12,529 | 13,299 | 5.79% |
| hubei | 847 | 14,183 | 15,030 | 5.64% |
| sichuan | 1141 | 19,376 | 20,517 | 5.56% |
| hunan | 831 | 14,373 | 15,204 | 5.47% |
| liaoning | 577 | 9967 | 10,544 | 5.47% |
| shanxi_1 | 506 | 9017 | 9523 | 5.31% |
| xizang | 28 | 597 | 625 | 3.48% |
| chongqing | 293 | 6690 | 6983 | 3.20% |
| Total | 26,845 | 369,789 | 396,634 | 6.77% |

Model 2 in Table 5 presents the regression results of Model (2). Model (2) mostly examines whether a significant difference exists in the interest rate of borrowing among different provinces under the control of other variables from the perspective of the borrower.

As shown in Model 2 in Table 5, the regression coefficients of most provinces are significant at the 5% level when other variables are controlled. The lower part of Table 5 shows the difference test results of the interest rates among the provinces and the difference between the borrowing rates of the provinces and the control group. The F values are 23.9152 and 23.3233, and both reject the null hypothesis. Significant differences exist among the regression coefficients of the provinces. The preceding results show that under the control of other variables, a significant difference exists in the interest rates of borrowing among different provinces. From the perspective of borrowers, a phenomenon of self-discrimination of borrowers in online lending exists. This result is similar to that of Mollick.

As shown by the regression results, the regression coefficients of the control variables are significant at the 1% level. In particular, for the control variable that provides the information of the order, the regression coefficient of the borrowing period is positive, thereby indicating that the borrowing period exhibits a positive correlation with the borrowing rate. When the borrowing period is long, the interest rate of borrowing is high. This result is due to the long-term loan. Short-term loans have high

risks and low liquidity, which require a high-risk premium to compensate and a high borrowing rate in the online lending market. The regression coefficient of the loan amount is significantly negative, thereby indicating that the loan amount is negatively correlated with the borrowing rate. When the borrowing amount is high, the interest rate of borrowing is low.

**Table 5.** Empirical test results of geographical discrimination.

| Variable | Model 1 | Model 2 | Variable | Model 1 | Model 2 | Variable | Model 1 | Model 2 |
|---|---|---|---|---|---|---|---|---|
| anhui | −0.0095 *** | 0.01 | jiangxi | −0.0149 *** | 0.051 | apr | −0.0039 *** | |
| | (−3.1592) | −0.3 | | (−5.9481) | −1.41 | | (−35.9534) | |
| beijing | −0.0145 *** | −0.2788 *** | liaoning | −0.0166 *** | 0.1178 *** | period | 0.0004 *** | 0.0236 *** |
| | (−6.8954) | (−9.0914) | | (−6.9647) | −3.39 | | −8.99 | −38.5 |
| fujian | −0.0108 *** | 0.037 | neimenggu | −0 | −0.2265 *** | ln_amount | −0.0323 *** | −0.0732 *** |
| | (−5.8263) | −1.37 | | (−1.1900) | (−5.6313) | | (−95.0217) | (−13.7813) |
| gansu | 5 × 10⁻⁴ | −0.1974 *** | ningxia | −0 | −0.02 | creditrating | −0.5304 *** | 0.8385 *** |
| | −0.15 | (−3.8841) | | (−0.7810) | (−0.2434) | | (−358.1010) | −38.9 |
| guangdong | −0.0052 *** | −0.1389 *** | qinghai | −0 | −0.11 | age | 0.0024 *** | 0.0264 *** |
| | (−3.4678) | (−6.3376) | | (−0.6700) | (−1.0703) | | −41.4 | −31.3 |
| guangxi | −0.0062 *** | −0.05 | shandong | −0.0091 *** | 0.2162 *** | marry | 0.0086 *** | −0.2041 *** |
| | (−2.6875) | (−1.3509) | | (−5.1799) | −8.47 | | −11.4 | (−18.4568) |
| guizhou | −0 | −0.2212 *** | shanxi1 | −0.0152 *** | −0.1496 *** | education | 0.0129 *** | −0.2160 *** |
| | (−0.7141) | (−5.4033) | | (−6.1322) | (−3.1393) | | −29 | (−33.3471) |
| hainan | −0.0070 * | −0 | shanxi2 | −0 | −0.0877 ** | company | 0.0113 *** | −0.0155 *** |
| | (−1.7753) | (−0.0450) | | (−1.3225) | (−2.3382) | | −33.3 | (−3.1479) |
| hebei | −0.01125 *** | 0.0645 ** | shanghai | −0.0155 *** | −0.1783 *** | income | 0.0164 *** | 0.0812 *** |
| | (−5.1279) | −2.02 | | (−7.0325) | (−5.5316) | | −46.8 | −15.8 |
| henan | −0.0038 * | −0.1957 *** | sichuan | −0.0175 *** | −0.1175 *** | house | 0.0025 *** | −0.0640 *** |
| | (−1.8426) | (−6.5979) | | (−9.1186) | (−3.2112) | | −3.06 | (−5.3427) |
| heilongjiang | −0.0103 *** | −0.02 | tianjin | −0.0166 *** | 0.2303 *** | car | 0.0111 *** | −0.2124 *** |
| | (−3.9664) | (−0.5698) | | (−3.3611) | −3.15 | | −12.1 | (−15.7899) |
| hubei | −0.0188 *** | −0.1128 *** | xizang | −0.01 | 0.183 | houseloan | 0.0082 *** | −0.2151 *** |
| | (−8.9226) | (−3.6695) | | (−0.7916) | −1.49 | | −7.58 | (−13.5710) |
| hunan | −0.0201 *** | −0.0880 *** | xinjiang | −0.0066 * | −0.3229 *** | carloan | 9 × 10⁻⁴ | −0.1531 *** |
| | (−9.5477) | (−2.8718) | | (−1.9491) | (−6.5458) | | −0.56 | (−6.6257) |
| jilin | −0.0052 * | 0.1185 *** | yunnan | −0.0049 * | −0.0663 * | | | |
| | (−1.7060) | −2.67 | | (−1.9061) | (−1.7610) | | | |
| jiangsu | −0.0050 *** | 0.0969 *** | chongqing | −0.0238 *** | 0.023 | | | |
| | (−2.7389) | −3.66 | | (−8.5419) | −0.56 | | | |
| R² | 0.3081 | 0.0201 | Adj R² | 0.3080 | 0.0200 | N | 396,634 | 396,634 |

Note: (1) Standard deviation and error are enclosed in parentheses; *, **, and *** denote the levels of significance, 10%, 5%, and 1%, respectively. (2) Stata software was used for the estimation in this study.

On the basis of the empirical results of geographical discrimination from the perspective of borrowers and lenders, significant differences exist in the success rates of borrowing among banks and the interest rate of borrowing; that is, significant geographical discrimination exists in online lending.

*4.3. Exploration of Potential Causes of Geographical Discrimination*

The test results show that significant geographical discrimination exists in online lending from the perspective of borrowers and lenders. Subsequently, this study will also analyze the causes of geographical discrimination in online lending from four aspects: economy, finance, education, and ethnicity. That is, this study will determine what type of regional differences cause the differences between the success and interest rates of borrowing among regions. This research uses the GDP of each province to measure economic differences among regions, the local budgetary general expenditures to measure financial differences, and education funds to measure educational differences. Minority autonomous regions and other provincial administrative units measure ethnic differences among regions. These variables were regressed by Models (3)–(10), and the following test results were obtained.

Table 6 shows the regression results of Models (3)–(6), which are primarily used from the perspective of lenders to study the impacts of economic, fiscal, educational, and ethnic differences between regions on the success rate of borrowing for online lending after controlling for other influencing factors.

From the perspective of economic disparity, the GDP level of a borrower's province exhibits a significant positive correlation with the success rate of borrowing in online lending. Provinces with higher GDP levels have higher rates of successful borrowing, indicating that the difference in the

rate of successful borrowing in online lending is affected by economic differences among provinces. The province's GDP level reflects the overall economic environment of the province. For provinces with a better economic environment, lenders often think that their income is more stable and their repayment ability is stronger. Therefore, lenders are inclined to provide loans to borrowers in provinces with higher GDP levels.

**Table 6.** Empirical analysis of the causes of regional discrimination (perspective of borrowers).

| Variable | Success | | | |
|---|---|---|---|---|
| prov_gdp | 0.0007 ** (2.4942) | – | – | – |
| prov_gov | – | 0.0009 *** (3.5838) | – | – |
| prov_edu | – | – | 0.0012 *** (3.5632) | – |
| prov_nat | – | – | – | 0.0044 *** (3.2280) |
| Control | – | – | – | – |
| R² | 0.3076 | 0.3076 | 0.3076 | 0.3076 |
| N | 396,634 | 396,634 | 396,634 | 396,634 |

Note: (1) Standard deviation and error are enclosed in parentheses; **, and *** denote the levels of significance, 5%, and 1%, respectively. (2) Stata software was used for the estimation in this study. (3) We have omitted the results of the control variables to ensure the legitimacy of the article.

From the perspective of fiscal differences, the local budgetary expenditure level of the borrower's province is significantly positively correlated with the success rate of borrowing. The higher the local budgetary expenditure level, the higher the success rate of borrowing, thereby indicating that the difference in the success rate of borrowing in online lending is affected by fiscal differences among provinces. For provinces with higher general budgetary expenditures for local finance, the government's finance is more secure, and lenders tend to lend money to borrowers in these regions, which leads to a high success rate for borrowing.

From the perspective of educational differences, the level of educational funding of a borrower's province is significantly positively correlated with the success rate of borrowing. The higher the level of education funds, the higher the success rate of borrowing in the province, thereby indicating that the difference in the success rate of borrowing among provinces varies. The level of educational funding of a province represents the education level among provinces to a certain extent. The education level frequently represents the overall quality of local residents. For borrowers in provinces with high education levels, the credit image of borrowers is relatively good. With all other things being equal, lenders are more willing to lend money to borrowers from these regions.

From the perspective of ethnic differences, the regression coefficient of ethnic minority autonomous regions is significantly positive. The success rate of borrowing in minority autonomous regions is significantly higher than that in other provincial administrative regions, thereby indicating that the success rate of borrowing among provinces is affected by ethnic differences, and lenders are more likely to lend money to borrowers from minority autonomous regions.

Table 7 provides the regression results of Models (7)–(10), which were mostly used from the perspective of borrowers to study the impacts of economic, fiscal, educational, and ethnic differences between regions on the borrowing rate of online lending after controlling for other influencing factors.

From the perspective of economic differences, the GDP level of a borrower's province is significantly positively correlated with the borrowing rate, thereby indicating that the difference in borrowing rates set by borrowers in online lending is affected by economic differences among provinces. The higher the GDP level of a province, the higher the borrowing rate set by the borrower. From the perspective of ethnic differences, the regression coefficient of minority autonomous regions is significantly negative, thereby indicating that the difference in borrowing rates set by borrowers in online lending is affected

by ethnic differences among provinces. The borrowing rates set by borrowers in minority autonomous regions are significantly lower than those in other provincial administrative regions. The fiscal and educational differences among the provinces exert no significant impacts on the interest rate set by the borrower.

**Table 7.** Empirical analysis of the causes of regional discrimination (perspective of lenders).

| Variable | Apr | | | |
|---|---|---|---|---|
| prov_gdp | 0.0111 *** (2.8052) | – | – | – |
| prov_gov | – | 0.0015 (0.3935) | – | – |
| prov_edu | – | – | 0.0012 (0.3124) | – |
| prov_nat | – | – | – | −0.0958 *** (−3.8638) |
| Control | – | – | – | – |
| $R^2$ | 0.0183 | 0.0183 | 0.0183 | 0.0184 |
| N | 396,634 | 396,634 | 396,634 | 396,634 |

Note: Annotation is the same as that in Table 6.

In summary, geographical discrimination exists in online lending. From the perspective of lenders, the provincial GDP level has a significant negative correlation with the success rate of borrowing. The provincial general budgetary expenditures and provincial levels of educational funding are significantly positively correlated with the success rate of borrowing. The success rate of borrowing in minority autonomous regions is significantly higher than that in other provincial administrative districts. These findings indicate that the success rate of borrowing is affected by the economic, financial, educational, and ethnic factors of the borrower's province. That is, a relationship exists between the lender's geographical discrimination against the borrower and the economic, fiscal, educational, and ethnic differences among provinces. From the borrower's point of view, the provincial GDP level is significantly positively correlated with the interest rate set by the borrower. The interest rate set by the borrowers in the minority autonomous regions is significantly lower than that in other provincial administrative regions. It shows that the borrowing interest rate is mainly affected by the economic and ethnic factors of the province. The influences of the economic and ethnic factors of the province where a person is located, i.e., the self-discrimination of the borrower, are primarily related to economic and ethnic differences among provinces.

*4.4. Robustness Test*

(1)　Transform sample group

This study first tests the robustness of the empirical results by transforming the sample set. It selects the loan order data of borrowers between the ages of 18 and 65 years as the sample. Subsequently, the age range of the borrower sample is narrowed down to 18–40 years to test the robustness of the empirical results. Table 8 presents the results of the robustness test for the explained variables as the success rate of borrowing. Table 9 shows the results of the robustness test for the explanatory variables as the borrowing rate. The regression coefficients of the main research variables exhibit no change in symbol and saliency, thereby indicating that the empirical results obtained in this study exhibit good robustness.

**Table 8.** Robustness test results of loan success rate (change sample).

| Variable | Success | | | |
|---|---|---|---|---|
| prov_gdp | 0.0006 ** (2.1622) | – | – | – |
| prov_gov | – | 0.0010 *** (3.8935) | – | – |
| prov_edu | – | – | 0.0013 *** (3.5583) | – |
| prov_nat | – | – | – | 0.0055 *** (3.9903) |
| Control | – | – | – | – |
| $R^2$ | 0.3020 | 0.3020 | 0.3020 | 0.3020 |
| N | 357,379 | 357,379 | 357,379 | 357,379 |

Note: (1) Annotation is the same as that in Table 6. (2) Use samples from age 18–40.

**Table 9.** Results of the robustness test of the borrowing rate (change sample).

| Variable | Apr | | | |
|---|---|---|---|---|
| prov_gdp | 0.0101 ** (2.4159) | – | – | – |
| prov_gov | – | 0.0005 (0.1175) | – | – |
| prov_edu | – | – | 0.0016 (0.3720) | – |
| prov_nat | – | – | – | −0.0987 *** (−3.7070) |
| Control | – | – | – | – |
| $R^2$ | 0.0199 | 0.0199 | 0.0199 | 0.0199 |
| N | 357,379 | 357,379 | 357,379 | 357,379 |

Note: (1) Annotation is the same as that in Table 6. (2) Use samples from age 18–40.

(2)   Using the probit model for inspection

The empirical results of this study are primarily based on the ordinary least squares (OLS) regression model. We choose the OLS model to regress the borrowing success rate because the regression coefficients of the dummy variables of each province represents its economic comparison with the control group or the difference in the success rate of borrowing in Zhejiang. However, the success of borrowing is a binary selection variable. Therefore, this study chooses the probit model, which is more suitable for studying binary selection variables for further robustness test, to eliminate the bias caused by model selection. Table 10 presents the regression results of the probit model. The sign and significance of the regression coefficients of the main research variables have not changed, thereby proving that our previous empirical research results are robust.

**Table 10.** Empirical analysis of the causes of regional discrimination (probit model).

| Variable | Success | | | |
|---|---|---|---|---|
| prov_gdp | 0.0133 *** (4.2342) | – | – | – |
| prov_gov | – | 0.0165 *** (5.5188) | – | – |
| prov_edu | – | – | 0.0192 *** (6.1319) | – |
| prov_nat | – | – | – | 0.0633 *** (3.1265) |
| Control | – | – | – | – |
| Pseudo $R^2$ | 0.3494 | 0.3495 | 0.3495 | 0.3494 |
| N | 396,634 | 396,634 | 396,634 | 396,634 |

Note: (1) Annotation is the same as that in Table 6. (2) Because the interest rate is not 0, the 1 variable cannot be used to test the robustness of the probit model.

In summary, the results of the robustness test obtained by transforming the sample set and the transformation measurement method show that the empirical test results obtained before this study are robust.

*4.5. Expansibility Test*

In the previous article, we analyzed the causes of regional discrimination from four aspects: regional economy, finance, education, and ethnic differences. In fact, economy, finance, education, and nationality are not completely separated. The influences of these factors should be considered from many angles. Since the regional economy is the core variable of the four aspects, we mainly use it to construct the interaction term. The reasons for choosing GDP as the core are: Firstly, we believe that online lending is an economic activity, so regional economic development is the most important. Secondly, in the study of regional differences in China, GDP is often used to measure the level of regional development (Hao et al. 2014).

Table 11 provides the regression results of the interaction effects of GDP and other causes. From the regression results, gdp_gov and gdp_edu of the borrower's province are positively related to the success rate and interest rate of the borrowing. This is consistent with the results in Tables 6 and 7: that provinces with higher GDP levels and higher levels of general fiscal expenditure and education expenditure have higher borrowing success rates and higher borrowing rates. This is the same as China's actual situation. Provinces with higher GDP, fiscal expenditure level, and education level indicate that borrowers have higher economic benefits, are easier to loan to, and can bear higher borrowing rates. gdp_nat is negatively correlated with success and positively correlated with apr. In Table 6, prov_nat is positively correlated with success, and prov_nat is negatively correlated with apr. It also shows to some extent that ethnic differences are also the cause of regional discrimination. It further explained that the lender's geographic discrimination against the borrower and the borrower's own regional discrimination are related to economic, fiscal, educational, and ethnic differences between provinces.

**Table 11.** Interactive effects of potential causes of geographic discrimination.

| Variable | Success | | | Apr | | |
|---|---|---|---|---|---|---|
| prov_gdp | 0.0007 (1.0532) | −0.0012 ** (−2.1153) | 0.0011 *** (3.7561) | 0.0486 *** (5.1687) | 0.0484 *** (5.7945) | 0.0045 (1.0734) |
| prov_gov | 0.0016 *** (3.0241) | – | – | −0.0395 *** (−4.6597) | – | – |
| prov_edu | – | 0.0035 *** (6.1846) | – | – | −0.0361 *** (−4.2888) | – |
| prov_nat | – | – | 0.0048 * (1.9111) | – | – | −0.0435 (−1.2570) |
| gdp_gov | 0.0017 *** (7.1994) | – | – | 0.0002 (0.0451) | – | – |
| gdp_edu | – | 0.0016 *** (6.7892) | – | – | 0.0064 * (1.8670) | – |
| gdp_nat | – | – | −0.0008 (−0.5586) | – | – | 0.0307 (1.5611) |
| Control | – | – | – | – | – | – |
| Pseudo R$^2$ | 0.3077 | 0.3077 | 0.3076 | 0.0184 | 0.0184 | 0.0184 |
| N | 396,634 | 396,634 | 396,634 | 396,634 | 396,634 | 396,634 |

Note: (1) Annotation is the same as that in Table 6. (2) gdp_gov is the interaction term between GDP level and local finance, gdp_edu is the interaction term between GDP level and education funding level, and gdp_nat is the interaction term between GDP level and whether it is a minority. (3) The interaction term interacts after the variables are centralized.

## 5. Conclusions and Reflections

As a financial model that has emerged with the development of Internet finance, online lending has experienced vigorous development in China in recent years. An increasing number of scholars have begun to pay attention to discrimination in the online lending market. On the basis of the loan

order data of China's earliest P2P network lending platform, this study analyzes whether geographical discrimination exists in online lending from two perspectives. Furthermore, this study identifies the reasons for the development of geographical discrimination in the online lending market from four aspects: economy, finance, education, and ethnicity. That is, we examine what types of regional differences cause the difference in the success and interest rates of borrowing among regions. This study uses the GDP of each province to measure the economic differences among regions, the local budgetary general expenditures to measure financial differences, and education funds to measure educational differences. Minority autonomous regions and other provincial administrative units measure ethnic differences among regions. The results of the study indicate the following.

First, significant geographical discrimination exists in China's online lending market. From the perspective of the lender, different investment intentions of borrowers from various regions are demonstrated, which leads to differences in the success rates of borrowings in different regions. From the perspective of the borrower, a belief exists that borrowers in different regions have varying borrowing rates because of the effect of geographical discrimination.

Second, the lender's geographical discrimination against the borrower is related to economic, fiscal, educational, and ethnic differences among provinces. From the perspective of lenders, the provincial GDP level has a significant positive correlation with the success rate of borrowing. The provincial general budgetary expenditures and provincial level of educational funding are significantly positively correlated with the success rate of borrowing. The success rate of borrowing in minority autonomous regions is significantly higher than that in other provincial administrative districts. These findings indicate that the success rate of borrowing is affected by the economic, financial, educational, and ethnic factors of the borrower's province. That is, a relationship exists between the lender's geographical discrimination against the borrower and the economic, fiscal, educational, and ethnic differences among provinces.

Finally, the borrower's self-discrimination is primarily related to economic and ethnic differences among provinces. From the borrower's point of view, the provincial GDP level is significantly positively correlated with the interest rate set by the borrower. The interest rate set by the borrowers in the minority autonomous regions is significantly lower than that in other provincial administrative regions, thereby indicating that the borrowing interest rate is mostly borrowed. The influences of the economic and ethnic factors of the province where a person is located, i.e., the self-discrimination of the borrower, are primarily related to economic and ethnic differences among provinces.

This study enriches research on Internet lending in the Internet financial market, supplements and achieves relevant research conclusions on the phenomenon of geographical discrimination in online lending, and exhibits certain reference significance for participants and regulators of the online lending market (Barrell and Nahhas 2019). In addition, this article also thanks the reviewers for putting forward some new thoughts on this article; for example, we can further consider the geographical distance between economic centers of gravity, and we will explore this in the future.

**Author Contributions:** All the authors have contributed to the whole development of the manuscript: designing the research, performing the calculations, writing the text, discussing the results, and obtaining the conclusions. All authors have read and agreed to the published version of the manuscript.

**Funding:** This paper was supported by the Key project of Chinese National Social Science Fund (Research on the Collaborative Path of the Evolution of Internet Lending and the Financing Advancement of Small and Micro Enterprises under the New Financial Normality, number 15AJY018) and the key project of Chongqing University (numbers CQDXWL-2014-Z019 and 106112016CDJXY020013).

**Acknowledgments:** Data from the Aijie database.

**Conflicts of Interest:** The authors declare no conflict of interest.

## Appendix A

**Table A1.** Variable Pearson correlation coefficient table.

| | Success | Apr | Period | Ln_amoun | Creditrating | Age | Marry | Education | Company | Income | House | Car | Houseloan | Carloan | Prov_gdp | Prov_gov | Prov_edu | Prov_nat |
|---|---|---|---|---|---|---|---|---|---|---|---|---|---|---|---|---|---|---|
| success | 1 | | | | | | | | | | | | | | | | | |
| apr | −0.090 ** | 1 | | | | | | | | | | | | | | | | |
| period | −0.101 ** | 0.061 ** | 1 | | | | | | | | | | | | | | | |
| ln_amoun | −0.127 ** | 0.010 ** | 0.472 ** | 1 | | | | | | | | | | | | | | |
| creditrating | −0.529 ** | 0.076 ** | 0.083 ** | 0.047 ** | 1 | | | | | | | | | | | | | |
| age | 0.108 ** | 0.028 ** | 0.030 ** | 0.231 ** | −0.109 ** | 1 | | | | | | | | | | | | |
| marry | 0.075 ** | −0.024 ** | 0.038 ** | 0.177 ** | −0.086 ** | 0.394 ** | 1 | | | | | | | | | | | |
| education | 0.107 ** | −0.071 ** | 0.022 ** | 0.093 ** | −0.125 ** | 0.036 ** | −0.028 ** | 1 | | | | | | | | | | |
| company | 0.069 ** | −0.016 ** | 0.065 ** | −0.084 ** | −0.052 ** | −0.053 ** | −0.084 ** | 0.177 ** | 1 | | | | | | | | | |
| income | 0.076 ** | 0.002 | 0.012 ** | 0.427 ** | −0.104 ** | 0.259 ** | 0.199 ** | 0.063 ** | −0.244 ** | 1 | | | | | | | | |
| house | 0.074 ** | −0.031 ** | 0.028 ** | 0.160 ** | −0.090 ** | 0.301 ** | 0.316 ** | 0.107 ** | −0.015 ** | 0.162 ** | 1 | | | | | | | |
| car | 0.089 ** | −0.045 ** | −0.020 ** | 0.208 ** | −0.124 ** | 0.191 ** | 0.255 ** | 0.066 ** | −0.115 ** | 0.314 ** | 0.281 ** | 1 | | | | | | |
| houseloan | 0.070 ** | −0.043 ** | 0.031 ** | 0.124 ** | −0.091 ** | 0.117 ** | 0.146 ** | 0.149 ** | 0.006 ** | 0.141 ** | 0.454 ** | 0.165 ** | 1 | | | | | |
| carloan | 0.029 ** | −0.026 ** | 0.011 ** | 0.114 ** | −0.042 ** | 0.049 ** | 0.102 ** | 0.006 ** | −0.079 ** | 0.179 ** | 0.102 ** | 0.424 ** | 0.103 ** | 1 | | | | |
| prov_gdp | 0.013 ** | 0.007 ** | −0.033 ** | −0.015 ** | −0.014 ** | −0.020 ** | 0.015 ** | −0.046 ** | 0.013 ** | 0.067 ** | −0.063 ** | 0.013 ** | −0.041 ** | 0.019 ** | 1 | | | |
| prov_gov | 0.017 ** | 0.001 | −0.031 ** | −0.013 ** | −0.016 ** | −0.017 ** | 0.005 ** | −0.046 ** | 0.019 ** | 0.055 ** | −0.044 ** | 0.015 ** | −0.022 ** | 0.014 ** | 0.895 ** | 1 | | |
| prov_edu | 0.016 ** | 0.002 | −0.032 ** | −0.015 ** | −0.016 ** | −0.025 ** | 0.017 ** | −0.069 ** | 0.002 | 0.054 ** | −0.050 ** | 0.019 ** | −0.031 ** | 0.024 ** | 0.885 ** | 0.937 ** | 1 | |
| prov_nat | −0.001 | −0.008 ** | 0.021 ** | 0.012 ** | 0.003 | 0.014 ** | −0.005 ** | 0.009 ** | −0.019 ** | −0.041 ** | 0.018 ** | −0.001 | 0.027 ** | −0.013 ** | −0.315 ** | −0.250 ** | −0.328 ** | 1 |

Note: ** Significant correlation at 0.01 level (both sides).

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
