# Peer review of "Does Geographical Discrimination Exist in Online Lending in China: An Empirical Study Based on Chinese Loan Platform Renren"

_ijfs, doi:10.3390/ijfs8010015_

Round 1
Reviewer 1 Report
The paper is a solid contribution to the emerging research area commonly called P2P (online) lending. The main problem in such lending is the asymmetric information between borrowers and lenders and any additional information can be useful as a screening device. The findings are robust that geographic discrimination emerges as a sorting device for lenders to discern the quality of online borrowers.
My main concern is that some typos remained in the current manuscript, e.g. equation (1) and equation (2) on pp. 6. Subscripts and superscripts are sometimes missing, a careful combing is necessary to weed out any remaining typos.
Author Response
Dear Reviewer 1:
Thank you very much for your letter dated 04 Nov 2019 and the review reports. Based on your comment and request, we have modified the original.
At your suggestion, we have modified the formula on page 6 you said, and we have revised the problems that may be found later.
Finally, thank you for your support. Due to the approaching Chinese New Year and the reasons for the Wuhan virus, our changes may not be as good as you wish. If you have new opinions, please let us know.
Yours sincerely,
Tianlei Pi
E-mail: pitianlei@cqu.edu.cn
Reviewer 2 Report
Reviewer Comments on IJFS-617817
This paper examines whether there are various forms of discrimination in online lending in China. The Authors state that there are more thsn 900,000 such records in the Abstract, but the descriptive statistics show about one-third of it. The estimation technique is simple OLS, which is inadequate. The paper does not provide diagnostic tests or residual properties. One important problem with large datasets is that even very small differences in magnitude can become statistically significant. The size effects should also be discussed. Alternative functional forms for the model are also missing.
The use of the word "discrimination" in the paper might also be misleading. What would be the motive for that after adjusting for risk factors across the borrowers? Any negative significant results do not necessarily indicate discrimination as we know from the literature on wage differentials.
Author Response
Dear Reviewer 2:
Thank you very much for your letter dated 11 Jan 2020 and the review reports. Based on your comment and request, we have modified the original. We attached revised manuscript in the formats of both PDF and MS word for your approval.
Due to the time requirements of the editorial department, we tried our best to complete the modification in the shortest possible time. A revised manuscript with the correction sections marked was attached as the supplemental material for easy check. Should you have questions, please contact us without hesitation.
Some of your questions were answered below:
The review’s comment:
The Authors state that there are more thsn 900,000 such records in the Abstract, but the descriptive statistics show about one-third of it.
The authors’ answer:
Thanks to the reviewers for their careful attention, we have modified the abstract: Methods: This study used 396,634 order data (935,037 original order data ) from the Renren Loan website since its inception in January 2017. We used ordinary least squares (OLS) regression to study the problem of geographical discrimination in online lending in China, and we conducted two robust tests.In addition, we explained in article 3.1 Data source and processing: We processed the obtained original order data in the following aspects. First, order data disclosed by borrowers with incomplete information were excluded. Second,....... Finally, the number of valid data obtained was 396,634.
The review’s comment:
The use of the word "discrimination" in the paper might also be misleading.
The authors’ answer:
Thanks to the reviewer for reminding us, in fact, we considered this issue in the initial naming, but in the end we chose the word “discrimination”.
The term “discrimination” is also used in some of our references,such as
Pope D G, Sydnor J R. What’s in a Picture? Evidence of Discrimination from Prosper.com [J]. Journal of Human Resources,2011,46(1):53-92.
Chen Wei, Ye Dezhu. Research on Sex Discrimination in Internet Finance in China[J].Financial Review,2016(2):1-15.
Jiang You, Zhou Anqi. Is there any geographical discrimination in P2P network lending?——Experience data from “everyone's loan”[J].Journal of Central University of Finance and Economics,2016(9):29-39.
In addition, we also found words such as “difference”,” bias”:
Peng Hongfeng, Yang Liuming, Tan Xiaoyu. How Regional Differences Affect P2P Platform Lending Behavior——Based on Empirical Evidence of “Everyone's Lending”[J].Contemporary Economic Science,2016,38(5):21-33.
Lin M, Viswanathan S. Home Bias in Online Investments: An Empirical Study of an Online Crowdfunding Market[J]. Management Science,2015,62 (5):1393-1413.
Therefore, in this situation, we have not modified the article for the time being. If you think that it is still necessary to modify, please let us know and we will definitely modify.
The review’s comment:
The estimation technique is simple OLS, which is inadequate.
The authors’ answer:
Thanks to the reviewers for their suggestions, we have added a probit model to the robustness test 2 for testing.
The review’s comment:
The paper does not provide diagnostic tests or residual properties. One important problem with large datasets is that even very small differences in magnitude can become statistically significant. The size effects should also be discussed. Alternative functional forms for the model are also missing.
The authors’ answer:
Thanks to the reviewers for their suggestions, but unfortunately, due to the limitation of the modification time and model settings, we are temporarily unable to modify this part. If the reviewers have relevant literature provided, we are very willing to make further in-depth. I would like to express my apologies to the reviewers.
Finally, we sincerely thank the reviewers for their suggestions. At the same time, due to time and our understanding of English, we apologize for the insufficient amendments. I wish you a happy work, and I wish you a happy Chinese New Year as the Chinese New Year approaches.
Yours sincerely,
Tianlei Pi
E-mail: pitianlei@cqu.edu.cn

Reviewer 3 Report
Summary:
Online lending has developed rapidly in China in recent years as a typical Internet financial model. China's online lending related issues have received widespread attention from scholars. Methods: This study used 935,037 order data from the Renren Loan website since its inception in January 2017. This paper used ordinary least squares (OLS) regression to study the problem of geographical discrimination in online lending in China, and the authors conducted two robust tests. Results: Studies have shown that significant geographical discrimination exists in China’s online lending market. From the perspective of the lender, different investment intentions exist for borrowers from various regions, thereby leading to variations in the success rate of borrowings. From the perspective of the borrower, the belief exists that borrowers from different regions will have varying interest rates because of the effect of geographical discrimination. Conclusion: The authors believe that geographical discrimination is due to the effects of the economic, financial, educational, and ethnic conditions of the borrower’s location on willingness to invest and the success rate of borrowing. However, borrower’s self-discrimination is primarily related to economic and ethnic differences among provinces.
Assessment:
This is about a very important topic. The author(s) seem in control of the relevant econometric techniques (the general analysis is appropriate). The author(s) certainly dedicated sufficient time in understanding the dynamic of the system, run the appropriate tests. It is clear the contribution to the literature and the innovation provided by the authors of this paper.
I spotted some deficiencies which hopefully can be rectified. Below are my observations, which I would like to share with the author(s).
In terms of the cited references, please provide the publication year for each cited paper in the text. It is not clear what type of GDP the author(s) used. It is better to link between the results and the literature review. One recent paper Barrell and Nahhas who run an exercise based on the BIS for a much larger panel and longer period. The paper could be a good reference to help structure your paper. ( especially in terms of distance and interest rate).Reference:
Barrell, R., & Nahhas, A. (2019). The role of lender country factors in cross border bank lending. International Review of Financial Analysis. , https://doi.org/10.1016/j.irfa.2019.01.008
Best wishes and good luck!
Author Response
Dear Reviewer 3:
Thank you very much for your letter dated 13 Jan 2020 and the review reports. Based on your comment and request, we have modified the original. We attached revised manuscript in the formats of both PDF and MS word for your approval.
Due to the time requirements of the editorial department, we tried our best to complete the modification in the shortest possible time. A revised manuscript with the correction sections marked was attached as the supplemental material for easy check. Should you have questions, please contact us without hesitation.
Some of your questions were answered below:
The review’s comment:
In terms of the cited references, please provide the publication year for each cited paper in the text.
The authors’ answer:
Thanks to the reviewers for their careful attention, we have increased the year value of missing year documents.
The review’s comment:
One recent paper Barrell and Nahhas who run an exercise based on the BIS for a much larger panel and longer period. The paper could be a good reference to help structure your paper. ( especially in terms of distance and interest rate).
The authors’ answer:
Thanks to the reviewers for their valuable comments, we downloaded this article, read it, and added it to our references. That paper use country-level (consolidated) data available from the Bank for International Settlements (BIS), and an extended model based on home and host country characteristics conditioned on distance and mass primarily measured by GDP is used to explain the behaviour of cross-border lending stocks.
That article gave us a lot of thought by using the geographical distance between economic centers of gravity as a reference, but it is limited by the quantification of data, the measurement of the distance of China's online lending areas, and the setting of models. We will conduct it in the future. The outlook for this part is explained in our conclusions.
The review’s comment:
It is not clear what type of GDP the author(s) used.
The authors’ answer:
Thanks to the reviewers for their comments. In fact, the GDP selected in this article is from the National Bureau of Statistics of China. According to China's current statistical system, the GDP released by the National Bureau of Statistics of China is calculated based on the production method. The value of goods and services created in the process is a method to obtain added value by excluding the value of intermediate goods and services invested in the production process. The formula for calculating the added value of the production law of various industries in the national economy is as follows: added value = total output-intermediate input. Adding the added value of the production laws of various industries in the national economy, we get the GDP of France. Then it is divided into 5 groups according to the level of GDP, and each group is assigned a value of 1-5 from low to high.
Finally, we sincerely thank the reviewers for their suggestions. At the same time, due to time and our understanding of English, we apologize for the insufficient amendments. I wish you a happy work, and I wish you a happy Chinese New Year as the Chinese New Year approaches.
Yours sincerely,
Tianlei Pi
E-mail: pitianlei@cqu.edu.cn